# The Difference in Structural States between Canonical Proteins and Their Isoforms Established by Proteome-Wide Bioinformatics Analysis

**DOI:** 10.3390/biom12111610

**Published:** 2022-11-01

**Authors:** Zarifa Osmanli, Theo Falgarone, Turkan Samadova, Gudrun Aldrian, Jeremy Leclercq, Ilham Shahmuradov, Andrey V. Kajava

**Affiliations:** 1CRBM, Université de Montpellier, CNRS, 1919 Route de Mende, CEDEX 5, 34293 Montpellier, France; 2Institute of Biophysics, ANAS, Baku AZ1141, Azerbaijan

**Keywords:** isoform, large-scale analysis, protein structure, AlphaFold, canonical protein

## Abstract

Alternative splicing is an important means of generating the protein diversity necessary for cellular functions. Hence, there is a growing interest in assessing the structural and functional impact of alternative protein isoforms. Typically, experimental studies are used to determine the structures of the canonical proteins ignoring the other isoforms. Therefore, there is still a large gap between abundant sequence information and meager structural data on these isoforms. During the last decade, significant progress has been achieved in the development of bioinformatics tools for structural and functional annotations of proteins. Moreover, the appearance of the AlphaFold program opened up the possibility to model a large number of high-confidence structures of the isoforms. In this study, using state-of-the-art tools, we performed in silico analysis of 58 eukaryotic proteomes. The evaluated structural states included structured domains, intrinsically disordered regions, aggregation-prone regions, and tandem repeats. Among other things, we found that the isoforms have fewer signal peptides, transmembrane regions, or tandem repeat regions in comparison with their canonical counterparts. This could change protein function and/or cellular localization. The AlphaFold modeling demonstrated that frequently isoforms, having differences with the canonical sequences, still can fold in similar structures though with significant structural rearrangements which can lead to changes of their functions. Based on the modeling, we suggested classification of the structural differences between canonical proteins and isoforms. Altogether, we can conclude that a majority of isoforms, similarly to the canonical proteins are under selective pressure for the functional roles.

## 1. Introduction

Alternative splicing is one of the principal sources of structural and functional diversity in the proteomes of multicellular organisms. It is a process that may include or exclude particular exons of a multi-exonic gene from its processed messenger RNA. Different combinations of exons can produce multiple mRNA isoforms of a single gene. It is estimated that up to 95% of human multi-exonic genes are alternatively spliced [1,2]. The average number of splice variants per human gene is equal to four [3]. All this can drastically increase the number of different proteins in the proteome. Today, most genome-wide information about alternative splicing is generated on the nucleic acid level thanks to high-throughput data such as expressed sequence tags (ESTs) [4], microarrays [5], and RNA-seq data [6]. However, not all splicing variants are expressed as functional proteins. Although a very large number of alternatively spliced variants are detected in RNA-seq studies, large-scale mass spectrometry-based proteomics analyses detect only a small fraction of alternative isoforms on the protein level [7]. One of today’s problems in this area is to establish the real number of splice variants that appear as functional proteins for each gene. In addition to the application of genome-wide mass spectrometry analyses, researchers pay special attention to the protein isoforms with the most cross-species conservation and those that are able to maintain protein structure integrity [1,8,9,10].

Although the way to obtain the exact set of real protein variants may take some time, the data already available thanks to a combination of approaches (proteomics, cross-species conservation, and 3D mapping) can be used for the subsequent structural and functional annotations. Today, high-quality collections of protein isoforms are stored in UniProt, NCBI RefSeq, Ensembl databanks [11,12,13], and in more specific ones such as APPRIS, ISOexpresso, and ASES [14,15,16].

Another important point is the existence of a single main protein isoform among several protein variants for each gene, which is called principal isoform or canonical protein. The canonical protein is identified by several criteria: experimental data on its functional role; data about its expression in different tissues of an organism; existence of the same combination of exons in orthologous proteins and in different curated databases. Although, in the annotated databases of proteomes [11,12,13] many canonical proteins are well distinguished from their isoforms, some of them are still poorly annotated.

Depending on the proteomes and quality of their annotation, the number of isoforms usually exceeds the amount of canonical proteins 2–3 times [11,17]. At the same time, if to compare the number of proteins with the available experimental structural information, the situation is opposite. Almost all proteins in the Protein Data Bank [18] are canonical. Thus, due to a large gap between abundant sequence information and meager structural data on the isoforms, there is a growing interest in assessing the structural states and functional roles of alternative protein isoforms. As we have already mentioned, the sequence data on the isoforms are abundant. Therefore, if we want to get a global view of the structural-functional difference between the canonical proteins and their isoforms, apparently, the most appropriate approach is bioinformatics rather than the time-consuming experimental methods. In line with this need, during the last decade, significant progress has been achieved in the development of bioinformatics tools for large-scale structural and functional annotations of proteins. In the early days of structural bioinformatics, the foremost efforts of researchers were devoted to proteins with globular 3D structures. However, today, it is becoming clear that non-globular protein regions, having either intrinsically disordered conformations, membrane domains, elongated structures with tandem repeats or being aggregation-prone also have important functional roles [19,20,21]. Thus, an accurate structural and functional prediction of protein molecule can only be achieved when accounting for all these structural states. Recently, in line with this need, we developed a computational pipeline called TAPASS, which was designed to do just that [20]. The TAPASS pipeline is using known cutting-edge predictors able to detect intrinsically disordered regions (IDRs), transmembrane regions, signal peptides, conserved structured domains, short linear motifs (SLiMs) and aggregation-prone regions in protein sequences. The main novelty of this tool is a more precise prediction of aggregation-prone regions by taking into consideration the other known or predicted structural states. Moreover, the appearance of the AlphaFold program [22] opened up the possibility to model a large number of high-confidence structures of the isoforms. This artificial intelligence program, in a short time, became the gold standard computational technique for prediction of the 3D structure of proteins based on their sequence thanks to its accuracy competitive with experimental structures in a majority of cases.

In this study, by taking advantage of these state-of-the-art bioinformatics tools, we systematically compared the structural states of canonical proteins and isoforms. The analysis was performed on a large scale using 58 eukaryotic proteomes and provided a global view of the prevalence of each of these types of structures in canonical and isoform sets. Moreover, in some cases, our analysis proposed functional implications caused by structural changes of the isoforms as well as the possibility of selective evolutionary pressure, to which they can be exposed for functional roles.

## 2. Materials and Methods

### 2.1. Construction of Datasets of Canonical Proteins and Their Isoforms

#### 2.1.1. Main Dataset

Construction of properly divided large datasets of canonical proteins and their isoforms represents a challenge because some proteins are still poorly annotated. To obtain large subsets of canonical proteins and their isoforms, we retrieved corresponding sequences from reference proteomes of 58 eukaryotic species (Appendix A) by using July 2020 release of UniProt databank [11]. Our choice was justified by the fact that UniProt contains a large combined set of several databases. The UniProt uses the following criteria to identify the canonical proteins: experimental data on their functional role; data about their expression in different tissues of an organism; existence of the same combination of exons in orthologous proteins and in different curated databases (https://www.uniprot.org/help/canonical_and_isoforms (accessed on 25 August 2020)). First, we used option “Download all (FASTA (canonical & isoform)” to get 1,906,397 sequences including both canonical proteins and their isoforms. Second, we used “Download one protein sequence per gene” option to obtain a better-defined set of 1,244,044 canonical proteins. To avoid redundancy, we clustered the isoforms by CDhit [23] and removed the identical ones. This gave us 661,745 isoforms. Then we selected those isoform sequences, which had the same gene IDs as proteins from the canonical set and were highly similar BLAST (e-value < 10^−35^) with them [24]. As a result, we obtained a dataset of 263,475 canonical proteins and 565 942 isoforms, which was used in our analysis (Appendix A).

#### 2.1.2. Dataset of Proteins from Cancer-Related Genes with Well-Documented Expression Levels

Not all proteins from the UniProt databank have information about their expression level. Therefore, we built a smaller set of canonical proteins and corresponding isoforms of human cancer-related genes with well-documented expression levels in both 22 normal and cancer tissues. For this purpose, we used ISOexpresso database [15]. Our dataset contains 82 canonical and 166 isoform proteins, which were used for evaluation of the correlation between aggregation and expression level of proteins.

#### 2.1.3. Datasets for Estimation of the Structural Difference in Isoforms by Using AlphaFold Modeling

To evaluate the structural changes caused by the differences in the sequences (hereafter referred to as difference regions) of the corresponding canonical and isoform proteins, we used pairs of proteins with the difference regions inside well-conserved structured domains. For this purpose, we chose human proteins annotated in SwissProt [25] and having evidence of existence at the protein level (PE = 1). The conserved structural domains were detected by using HMM library of the CATH databank [26]. In the next step, we selected CATH domains that overlapped with the difference regions. A CATH domain found in a canonical protein may be shortened in the isoform so that the remaining domain is not able to fold. Therefore, we considered only isoforms where (1) the canonical CATH domain is shorter than 200 aa, and at least 70% of the domain remains in the isoform, or (2) the canonical domain is longer than 200 aa, and at least 50% of the domain remains in the isoform. For the modeling, we subsequently selected 168 canonical proteins with 223 corresponding isoforms where the difference regions were longer than 20 AA and located inside the CATH domains. Finally, to select the most conserved and studied domains, we ran the 168 canonical proteins by local BLASTP against PDB sequences from 7 species (*P. troglodytes*, *B. taurus*, *M. musculus*, *R. norvegicus*, *D. rerio*, *D. melanogaster*, *C. elegans*) and kept only those having more than 10 hits with e-value < 10^−6^. As a result, we obtained 53 canonical human proteins with 63 corresponding isoforms for the prediction by the AlphaFold program.

Subsequently, the 3D structures of the isoforms were predicted by AlphaFold Colab [27]. The structural models of the canonical proteins were obtained from the AlphaFold database (https://alphafold.com/download#proteomes-section (accessed on 10 May 2022)). The obtained structural models were analyzed by using PyMol [28].

### 2.2. Bioinformatics Tools Used to Annotate Structural States of Proteins

To annotate the structural states of proteins, we used the TAPASS pipeline, which includes several prediction tools. Structured domains were predicted by using HMM libraries (e-value < 10^−3^) of CATH. Intrinsically disordered regions were detected by IUPred [29] and an in-house BISMM filter, which chooses hydrophilic regions greater than 75% and proline-rich regions more than 25%. Signal peptide and transmembrane regions were predicted with SignalP and TMHMM, respectively [30,31]. The tool also predicts amyloidogenic regions (aggregation-prone motifs) by ArchCandy2.0 [32], TANGO [33], and PASTA 2.0 [34] with their default parameters. We detected short linear motifs (SLiMs) of degradation (degrons) by using motifs collected in the Eukaryotic Linear Motif (ELM) resource [35].

### 2.3. Detection of Structural Changes in and around the Difference Regions

All types of difference regions (insertion, deletion, non-identical, and mixed) can cause structural changes not only in the place of their location but also in the flanking regions with identical sequences. Most of the methods used in the TAPASS for structural annotation of canonical and isoform proteins detected these changes automatically. However, cases when deletions truncated CATH domains required additional rules (see Section 2.1.3). The application of these rules in our analysis affected the prediction of structured/unstructured regions and exposed aggregation-prone regions (EARs).

### 2.4. Analysis of Tandem Repeats in Canonical Proteins and Isoforms

Tandem repeat regions were identified by MetaRepeatFinder (MRF) (https://bioinfo.crbm.cnrs.fr/index.php?route=tools&tool=15 (accessed on 6 July 2022)) [36] tool in five proteomes (*H. sapiens*, *M. musculus*, *D. melanogaster*, *D. rerio*, *A. thaliana*). From several tandem repeat finders of MRF, we chose Regex, T-REKS [37], and TRUST [38], which are specialized in the detection of TRs with units of less than 3 residues, less than 15 residues, and more than 15 residues, respectively. As a result, the combination of these finders detects all types of tandem repeats. The overlap between the “difference” region and the TR region was counted if they had at least one common residue.

## 3. Results and Discussion

### 3.1. Identification, Classification, and Distribution of Difference Regions

Difference in the sequences of canonical proteins and their isoforms is quite specific in comparison with the differences between orthologous/paralogous proteins. Frequently, the differences between the orthologues represent point mutations and (or) short indels spread over the proteins. While canonical proteins and their isoforms always have a region(s) with identical sequences (corresponding to the same exons) and relatively long fragments where sequences can be completely different (Figure 1). To detect the difference regions, we generated pairwise alignments of canonical-isoform proteins by using Clustal Omega [39] and treated them by our in-house script (Appendix A).

We classified the differences between the canonical-isoform pairs into four groups choosing as a starting point canonical sequence: insertion, deletion, non-identical and mixed (Figure 1). The “non-identical” regions have different sequences of the same length. “Mixed” regions are those that have both amino acid substitutions and indels in the difference region. Sometimes, these regions also include identical regions shorter than 10aa.

The analysis showed that the “mixed” difference region is the most common case, followed by the deletions (Figure 1B). At the same time, a more detailed analysis of the “mixed” cases showed that it also contains a significant amount of deletions (68.6% of positions have deletions, 15.4% insertions, and 16% amino acids). Because of the frequent deletions, on average, the isoforms are shorter in length than canonical proteins (Figure 1C).

### 3.2. Distribution of Structured and Unstructured Regions

Previous studies suggested that isoform proteins have a higher coverage of unstructured regions in comparison to canonical proteins [40,41,42]. This conclusion suggested a lower level of involvement of isoforms in functional activity than of canonical ones. We examined this conclusion by using our datasets and the TAPASS pipeline [20] (see Section 2.1.3). Our analysis showed that the proportion of proteins containing unstructured regions is slightly higher in the isoform set (Figure 2). The same tendency was observed when we compared the coverage of unstructured regions in proteins. At the same time, both of these differences were not statistically significant. Thus, our results do not confirm the previous conclusions about the higher number of unstructured residues in isoforms, rather suggesting that the canonical proteins and their isoforms have the same ratio of residues in structured/unstructured states. This also suggests that during evolution, isoforms preserve their structural domains, which play functional roles (Appendix A).

### 3.3. Changes in Subcellular Localization

To understand the functional role of a protein, it is important to know where it resides in the cell. There are a number of bioinformatics tools that can accurately predict the outcome of protein targeting in four major subcellular localizations: secreted proteins can be identified by SignalP [30], transmembrane regions (more exactly transmembrane helices) by TMHMM [31], nuclear proteins with nuclear localization signals can be found by regular expressions [35], and the remaining proteins as a rough approximation can be considered as cytosolic.

Our analysis of the proportion of proteins with signal peptide showed that it is significantly lower in isoforms than in canonical proteins (Figure 3A). It suggests that in some cases, the isoforms may maintain their globular functional domains but change their cellular localization from extracellular to cytosolic. A similar tendency was observed with the canonical proteins containing transmembrane helices (Figure 3B). Moreover, we found that the proportion of the nuclear localization signals in isoforms is significantly higher in comparison with canonical proteins. It indicates that isoforms are more often localized in the nucleus than canonical proteins (Figure 3C). The proportion of canonical proteins with transmembrane helices is higher than in isoforms, suggesting that a noticeable part of the isoforms loses their transmembrane localization. Parts of the difference regions that gain and lose signal peptides represent 2% and 4%, respectively. For the transmembrane helices, it is 2% and 7%. These changes may have important functional implications (Appendix A).

### 3.4. Proportion of Aggregation-Prone Regions

Proteins are usually soluble and easily degraded by proteases after having performed their functions. However, some of them, depending on the amino acid sequence and at certain conditions, can assemble into stable, protease-resistant aggregates. These aggregates are linked to serious diseases, which include, but are not limited to, Alzheimer’s disease, Parkinson’s disease, type II diabetes, and rheumatoid arthritis [43]. Moreover, protein aggregation can be “functional” and play a central role in liquid–liquid phase separation (LLPS), a process that leads to the formation of membrane-less organelles [44,45]. Several computational programs for the prediction of protein aggregation have been developed [46]. The most realistic evaluation of the aggregation potential requires the prediction of motifs located within unstructured regions and being aggregation-prone, which we call “Exposed Aggregation-prone Regions” (EARs) [20]. Here, we analyzed the EARs in canonical proteins and isoforms. Our interest in this analysis was also because, in general, canonical proteins have a higher level of cellular expression in comparison with their isoforms. It is reasonable to assume that to avoid aggregation, canonical proteins with a higher expression level may have a lower aggregation potential. The other reason for the higher aggregation potential of the isoforms may be the truncation of native globular domains and the unfolding of their remaining parts. For example, it was shown that the p53 isoform Δ133p53β, which is critical in promoting cancer activity, is regulated through an aggregation-dependent mechanism [41]. The analyses of the truncated DNA-binding domain of Δ133p53β suggest that its remaining part is most probably unfolded and contains the EARs.

We estimated an average aggregation potential of canonical proteins and isoforms by the proportion of EAR-containing proteins predicted by one of the predictors (ArchCandy, Pasta, Tango) in these two datasets. Our analysis revealed that the median value of proportion for isoforms with EARs is almost the same as for canonical proteins (Figure 4 and Appendix A).

Although it is accepted that canonical proteins have higher expression levels than the isoforms [7,47], most proteins from our main dataset do not have reliable information about their expression level. Therefore, we also analyzed smaller sets with 82 canonical and 166 isoform proteins of human cancer genes with well-documented expression levels in normal and cancer tissues (Appendix A). These sets were used for evaluation of the correlation between aggregation and expression level of the proteins. The results confirm that the average expression level of canonical proteins is significantly higher than that of their isoforms. We also compared the relationship between the expression level and aggregation potential of proteins in normal and cancer cells. The results of the analysis are shown in Figure 5. The expression of canonical proteins is higher in both normal and cancer cells. At the same time, the expression level of all proteins slightly decreases in cancer cells. We also found that the proteins with EARs are expressed less in both normal and cancer cells than the ones without EARs. These results are in agreement with the assumption that to avoid aggregation, proteins with a higher expression level may have a lower aggregation potential.

### 3.5. Canonical Proteins Have More Degradation Motifs Than Their Isoforms

The abundance of proteins in the cell mostly depends on the balance of two opposite processes: expression and degradation. In general, canonical proteins have a higher level of cellular expression in comparison with their isoforms. It was interesting to understand if there is any difference between these proteins in terms of their degradation. The experimental data on protein degradation is limited and controversial. We compared canonical and isoform proteins in silico by analyzing the occurrence of degron motifs by TAPASS [20]. The degrons are short linear motifs that increase the targeting of proteins for degradation [48,49]. We found that canonical proteins have a higher proportion of degrons in comparison to the isoforms and this difference is statistically significant (Figure 6 and Appendix A).

If the more frequent occurrence of degrons in the canonical proteins causes their higher degradation rate in comparison with the isoforms, this may decrease the difference in the abundance between canonical proteins and isoforms. In its turn, a similar level of abundance may explain almost the same proportion of the aggregation-prone proteins predicted (Figure 4) for the canonical and isoform sets.

### 3.6. Occurrence of Tandem Repeats in Canonical Proteins and Isoforms

Many protein sequences contain arrays of repeats that are adjacent to each other [50,51] tandem repeats (TRs). *Several authors* have proposed that TRs might have evolved by exon duplication and rearrangement [52,53]. Therefore, it was interesting to get insight into the difference between canonical proteins and isoforms in these particular regions. We detected TRs in five well-annotated proteomes (*H. sapiens*, *M. musculus*, *D. melanogaster*, *D. rerio*, *A. thaliana*) by using MetaRepeatFinder (MRF) (https://bioinfo.crbm.cnrs.fr/index.php?route=tools&tool=15 (accessed on 6 July 2022)). These proteomes contain a total of 44,357 canonical proteins. We found that a large part (43%) of them contains at least one TR region, and each TR-containing protein has, on average, about two TR regions. A comparison of the occurrence of the TR regions in canonical proteins and isoforms revealed that isoforms have fewer TR regions than canonical proteins (0.5 vs. 0.81 TR region per protein) (Figure 7A). It is especially noticeable for TRs with a repeat length of 4–10 residues (Figure 7B). Partially, the decrease in TRs in the isoforms can be explained by the fact that among the differences between canonical proteins and isoforms, we predominantly observed deletions (see Section 3.1). It was interesting to study the relationship between the location of the TRs and the difference regions. Our analysis showed that among the difference regions detected in the aligned pairs, a significant part (35%) overlaps with TRs.

### 3.7. Differences within the 3D Structures of Canonical Proteins and Isoforms Predicted by AlphaFold

Our proteome-wide analysis provides a global view of the canonical-isoform protein difference. At the same time, it is also interesting to investigate these changes from within the 3D structures down to the atomic details. In orthologous and paralogous proteins, the difference in the amino acid sequences of more than 30% of identity may guarantee the same structural fold [54]. However, the character of the differences between canonical and isoform sequences is quite specific. They are identical at the location of the same exons; however, in the places of alternative splicing, they can have completely different sequences. This “mosaic” arrangement may trigger significant structural and functional changes.

Given the fact that almost all proteins with experimentally determined 3D structures are canonical, the comparison requires molecular modeling of isoform structures. Previously, this type of modeling of the isoform structures and their comparison with the structures of the corresponding canonical proteins was described for some particular proteins [10]. Today, with the development of an artificial intelligence program called AlphaFold [22], the scientific community got an opportunity to build high-quality structural models on a large scale. Here, we applied the AlphaFold program to obtain structural models of the isoform proteins. It was especially interesting to examine cases when the difference regions between the isoform and canonical proteins are conserved in several organisms and located within well-conserved structured domains. For the modeling, we used human proteins. To evaluate the cross-species conservation, we used seven species from the Animal Kingdom (*P. troglodytes*, *B. taurus*, *M. musculus*, *R. norvegicus*, *D. rerio*, *D. melanogaster*, *C. elegans*). We considered that AlphaFold structural models are reliable when their level of confidence (pLDDT) was higher than 70%, they did not have disallowed backbone conformations, and the inside residues of the structure were predominantly apolar and did not have charged residues, which were not involved in the ionic bonds. The detection of unstructured regions was based on criteria used in TAPASS [20]. Several isoforms had difference regions outside of the well-conserved structured domains, while inside these domains, they were identical to each other. Each group of these isoforms was reduced to one representative case. As a result, we compared the 3D structures of 50 canonical human proteins with 51 structural models of the corresponding isoforms predicted by AlphaFold. This allowed us to classify the 3D structure transformations into four subgroups.

#### 3.7.1. Exon Deletions with the Preservation of the Overall Structure


*Proteins with tandem repeats*


Though most of the selected proteins have globular structures, non-globular structures built of tandem repeats were found in 26% (13 of 51) of the cases. In the analyzed proteins with the difference regions inside of the complete structure, the most frequent situation is the deletion of one repetitive unit. As a rule, these changes (also with any integer number of the repeats) do not cause serious structural perturbations (Figure 8A). These cases are observed in proteins with tandem repeats from Class III, IV, and V [51,54,55]. In a few cases, the difference regions do not have an integer number of repeats. This could lead to structural changes if this difference is located in the middle of the repetitive structure. However, the isoform models showed that the change in the loop size between the repeats preserves the integrity of the whole structure (Appendix A). In other such cases, these difference regions are located at the terminal parts of the repetitive domains with no effect on the overall structure (Appendix A). The described structural changes preserve the overall structure by creating patches of new surfaces that can lead to the modification of protein functions.


*Globular proteins*


Among 51 analyzed pairs, there are 20 globular structures, representing 38% of the cases, with the deletions of exons in the middle of the structure. In most of these cases, the deletion does not lead to critical structural transformations (Figure 8A). In some cases, it makes shorter loops preserving α-helices or β-strands; sometimes, it removes one or several transmembrane helices. At the same time, these deletions can lead to changes in the binding properties of the isoforms and (or) changes in the oligomerization states of the protein [56].

#### 3.7.2. Exon Substitutions That Preserve the 3D Structure

The other subgroup of four analyzed proteins (8% of the cases) is characterized by substitutions of exons. The size of the substituted exons is the same or almost the same, and the sequences of canonical and isoform variants are not identical but similar. AlphaFold suggests that the new exons of the isoforms fit the native structure. This does not change the overall structure but leads to local changes on the molecular surface. This can be a basis for the modification of protein functions [57] (Figure 8B).

#### 3.7.3. Deletion That Is Substituted in the Structure by Another Part of the Molecule

We observed 6 of 51 cases (12%) where an exon deletion in the isoform removes a region that is critical for the structural integrity of the globular domain. In the AlphaFold model of the isoform, this part of the structure is filled by a new fragment, which, in the canonical protein, belongs to an unstructured region. This suggests that to provide structural diversity, proteins may have two or more neighboring regions. One is in the structure, and another is unstructured. If the first region is deleted in the isoform, the second one can dock into the structure, preserve it, and modify the function. (Figure 8C)

#### 3.7.4. Deletions That Destabilize Structured Domains

We found eight cases (representing 16%) where exon deletions may destabilize the 3D structure of the isoforms. It mostly happened in large multi-domain proteins. We assigned these examples to a separate subgroup. In these structures, the domain, which may be destabilized by the deletion of a critical part, can be transformed into an unfolded linker connecting the other globular domains. Instead, in the canonical structure, these domains are connected by the structured domain (Figure 8D). In the case of canonical proteins with a single structured domain, the isoforms may represent intrinsically disordered proteins.

#### 3.7.5. Limitations of AlphaFold in the Interpretation of the Conformational Changes

Our analysis revealed some limitations of AlphaFold modeling of the isoforms. For example, it is the case when we try to distinguish between isoforms with exon deletions, which preserve the overall structure, from the ones that destabilize it. In most of the cases, we could not base our decisions on the confidence score pLDDT for the reason that even structures, which missed a large part of the domain, frequently had pLDDT scores higher than 70%. These borderline cases were classified based on our visual analysis. In general, AlphaFold had a tendency to build isoform models that are very close to the canonical structures but with missing parts corresponding to the deleted exons. One of these examples is shown in Figure 8A, where an isoform of the canonical 7-bladed beta-propeller of guanine nucleotide-binding protein subunit beta-3 has six repetitive units. AlphaFold model of the isoform is almost identical to the canonical structure but misses one blade leading to the structure with an open beta-propeller. However, the SwissModel structure made based on the known 6-bladed structure (PDB code 1E1A) represents a closed 6-bladed beta-propeller. Such ambiguous cases cannot be resolved without experimental studies.

## 4. Conclusions

We took advantage of the progress achieved in the development of bioinformatics tools for large-scale structural annotations of proteins and examined the structural differences between canonical proteins and their isoforms. It became possible thanks to the TAPASS pipeline, which uses several state-of-the-art programs for the prediction of structured domains, unstructured regions, transmembrane regions, and aggregation-prone motifs [20]. Moreover, the availability of the AlphaFold program [22] opened up the possibility of modeling a large number of isoform structures. Altogether, our in silico analysis of 58 eukaryotic proteomes supported the concept that the majority of isoforms, similarly to the canonical proteins, are under selective pressure for functional roles. We also found that the proportions of proteins with a signal peptide and transmembrane helices are lower in isoforms than in canonical proteins. This suggested that some isoforms lose their transmembrane or extracellular localization and, eventually, their functional roles. At the same time, we did not observe significant differences between canonical proteins and their isoforms in the occurrence of unstructured regions or aggregation-prone motifs. Our modeling of the isoform structures demonstrated that the AlphaFold program is perfectly suitable for investigations of the structural differences of splicing variants at atomic details. It was shown that frequently the isoform sequences being different from the canonical ones still can fold in similar structures. At the same time, the isoforms may have significant structural rearrangements, which can lead to changes in their functions. We suggested the classification of the structural differences in the isoforms, which preserves the overall structure of the canonical proteins.

## Figures and Tables

**Figure 1 biomolecules-12-01610-f001:**
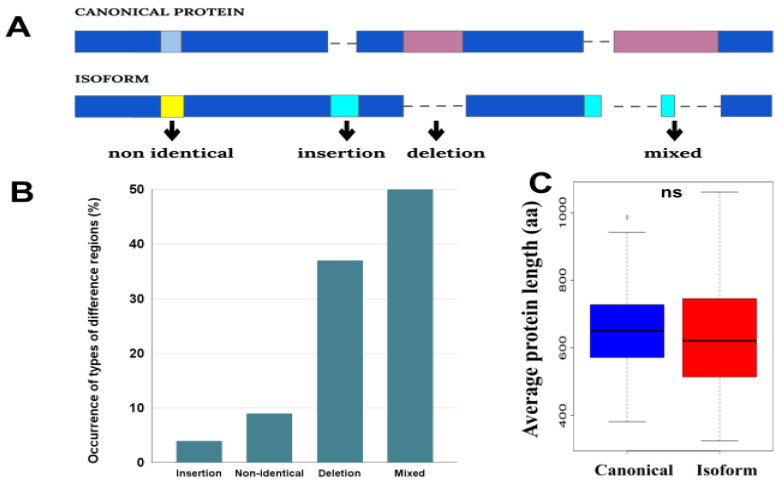
(**A**) Schematic representation of four groups of difference regions (dark blue and pink colors indicate identical and non-identical regions in the sequences, respectively). (**B**) Occurrence of types of the difference regions. (**C**) Distributions of the average length of canonical proteins and isoforms in proteomes. The distributions contain 58 points corresponding to the average length of each proteome. Here, ns means non-significant difference with *p*-value > 0.05.

**Figure 2 biomolecules-12-01610-f002:**
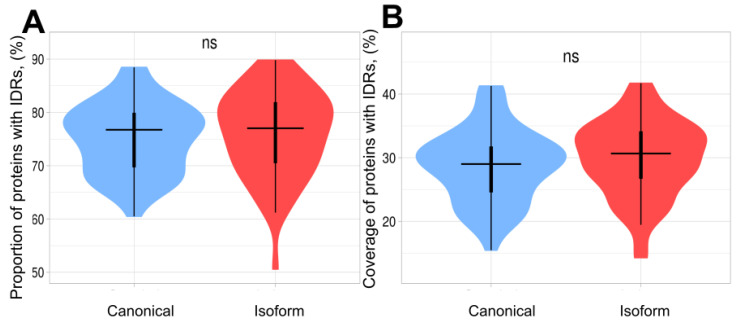
Violin plots of proportion and coverage of proteins containing IDRs in canonical and isoform proteins. The distributions contain 58 points corresponding to each proteome. (**A**) Proportion of proteins with IDRs in canonical proteins and isoforms. The difference between 2 sets is non-significant. (**B**) Coverage of IDRs in canonical proteins and isoforms. The coverage in isoforms is slightly higher; however, this difference is non-significant.

**Figure 3 biomolecules-12-01610-f003:**
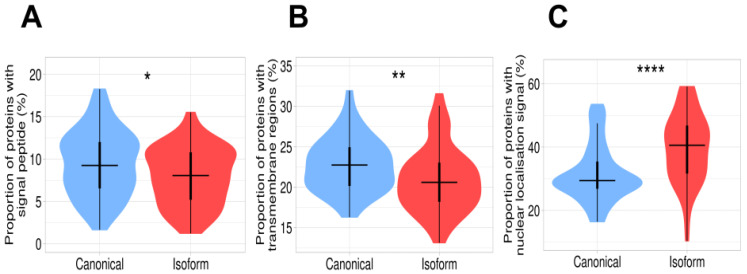
Difference in subcellular localization between canonical proteins and isoforms. (**A**) Proportion of proteins containing signal peptides. This value is significantly higher in canonical proteins than in isoforms. (**B**) Proportion of proteins containing transmembrane regions. The plot demonstrates a significant decrease in transmembrane proteins in the isoform set. (**C**) Proportion of proteins with nuclear localization signal. Isoforms have a remarkably high proportion of nuclear localization signals in comparison with canonical proteins. Signes *, **, **** mean significant differences with *p*-value < 0.05, *p*-value < 0.01, and *p*-value < 0.0001, respectively.

**Figure 4 biomolecules-12-01610-f004:**
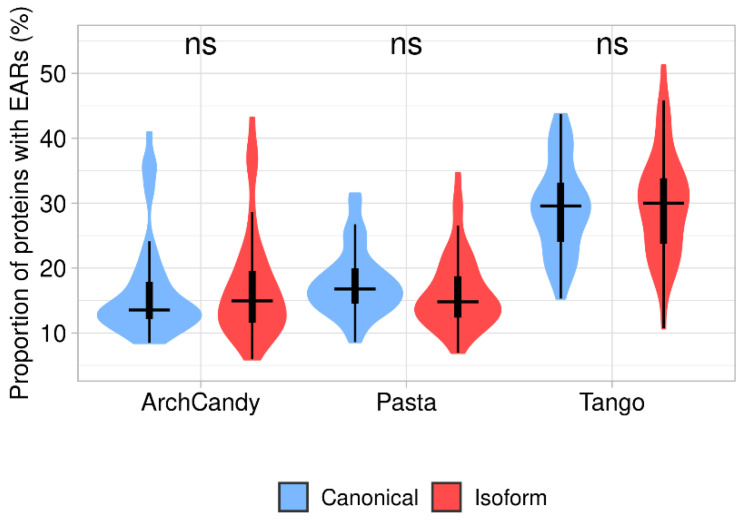
Proportion of EAR-containing proteins in canonical and isoform proteomes predicted by three tools (ArchCandy, Pasta, Tango). Differences between canonical proteins and isoforms are non-significant.

**Figure 5 biomolecules-12-01610-f005:**
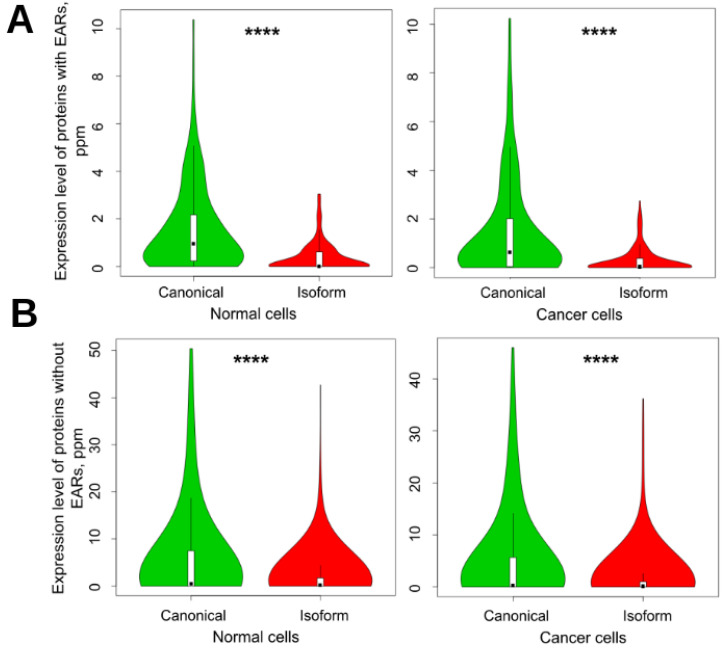
Violin plots of expression of canonical proteins and their isoforms in normal and cancer cells. (**A**) EAR-containing proteins and (**B**) non-EAR-containing proteins. EARs were predicted by using the ArchCandy program. Mean levels of expression for EAR-containing canonical proteins and isoforms in normal cells were 1.565 and 0.386, respectively, and in cancer cells, 1.490 and 0.306. For non-EAR-containing proteins, these values were 5.784, 1.773, and 4.984, 1.499, respectively. In accordance with *t*-test, all results were significant, with *p*-values of less than 10^−13^. **** means significant difference with *p*-value < 0.0001.

**Figure 6 biomolecules-12-01610-f006:**
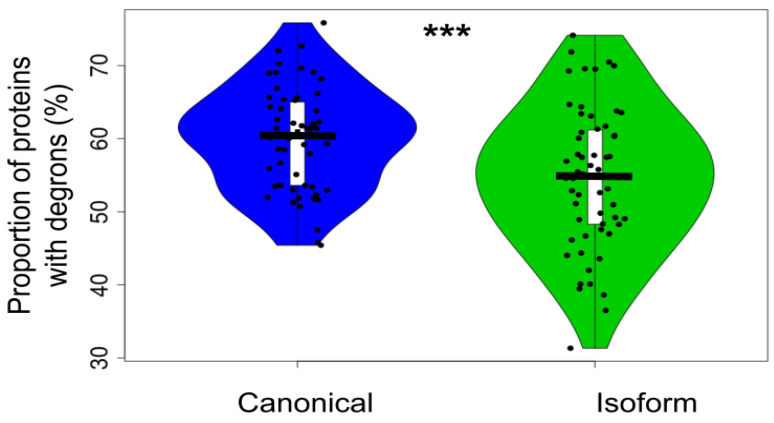
Proportion of canonical proteins and isoforms with degrons predicted by using SLiMs (*t*-test *p*-value = 0.00071). The distributions contain 58 points corresponding to each proteome. The proportion of degron-containing proteins is significantly higher in the canonical set than in the isoform one. Here, *** means significant difference with *p*-value < 0.001.

**Figure 7 biomolecules-12-01610-f007:**
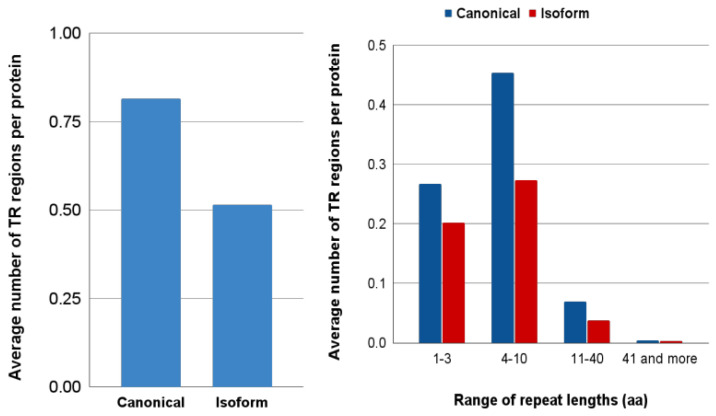
(**A**) Average number of tandem repeat regions determined per protein by MRF tool; (**B**) Distribution of proteins with tandem repeat by the length of their repetitive units.

**Figure 8 biomolecules-12-01610-f008:**
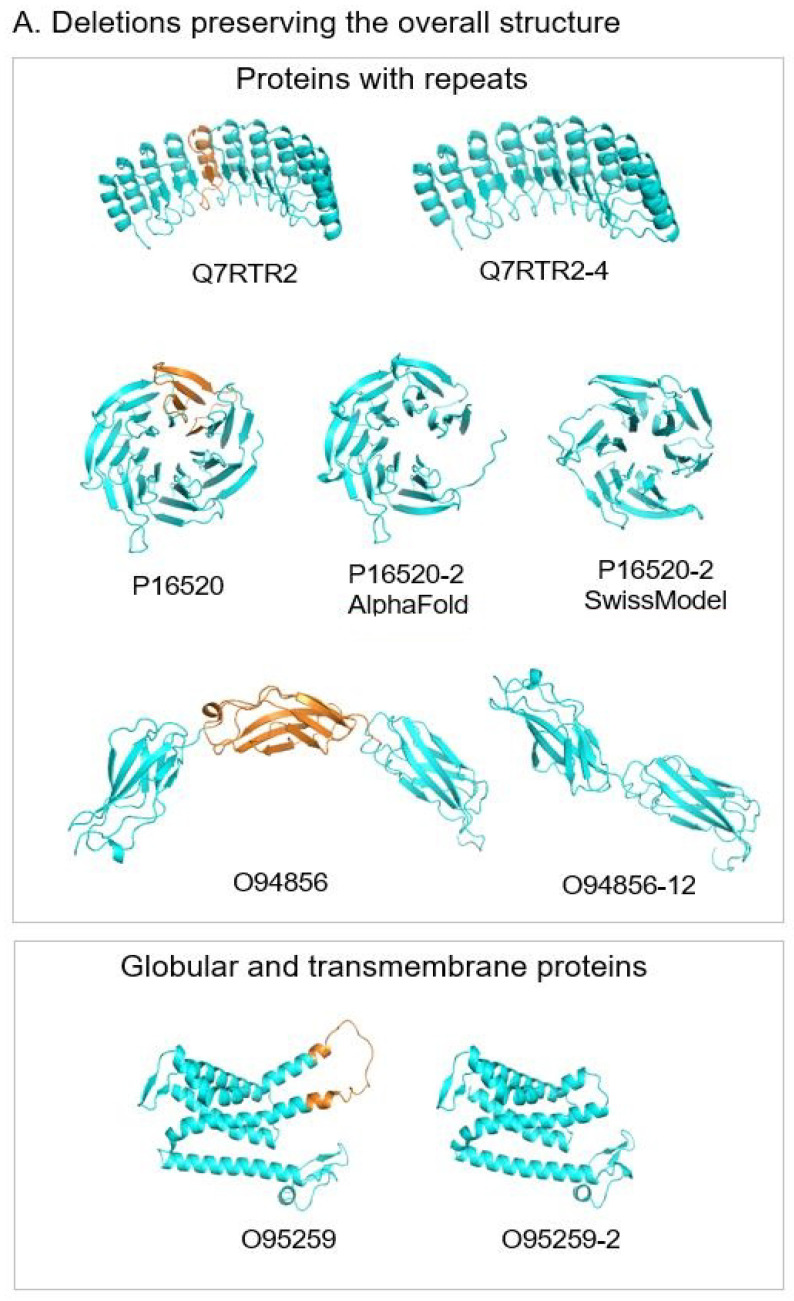
Ribbon representation of AlphaFold models of canonical proteins (left) and their isoforms (right). Fragments of canonical proteins deleted in the isoforms are in orange. Fragments of isoforms that substitute deleted fragments of the canonical proteins are in magenta. Representative structures of each subgroup from top to bottom are: (**A**). Deletions preserving the overall structure. Q7RTR2, LRR-protein of NLR family CARD domain-containing protein 3; P16520, 7-bladed beta-propeller of Guanine nucleotide-binding protein G(I)/G(S)/G(T) subunit beta-3. AlphaFold model of isoform represents 6-bladed structure with an open beta-propeller, SwissModel structure made based on the known 6-bladed structure (PDB code 1E1A) has closed beta-propeller; O94856, neurofascin; O95259, potassium voltage-gated channel subfamily H member 1; (**B**). Substitutions preserving the structure. P11362, fibroblast growth factor receptor 1; (**C**). Deletions replaced by another part of the protein. O00762, ubiquitin-conjugating enzyme E2 C, on the right, in yellow, the known crystal structure of ubiquitylation module similar to the truncated structure of the isoform in the center; (**D**). Deletions destabilizing structured domains. P13569, cystic fibrosis transmembrane conductance regulator.

## Data Availability

Not applicable.

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
