# Peer review of "The Difference in Structural States between Canonical Proteins and Their Isoforms Established by Proteome-Wide Bioinformatics Analysis"

_biomolecules, 2022, doi:10.3390/biom12111610_

Round 1

Reviewer 1 Report

Using bioinformatics methods, the authors systematically compared structural states of canonical proteins and their isoforms arising due to alternative splicing. The structural states included structured domains, intrinsically disordered regions, aggregation-prone regions and tandem repeats. They found that the isoforms have less signal peptides, transmembrane regions or tandem repeat regions in comparison with their canonical counterparts.

My comments:

It would be useful to give a short description of the AlphaFold and TAPASS programs.

It would be good to describe the Figures 2, 3, 4, and 6 in more detail.

The text contains only one reference to the work by Prof. Uversky, while the manuscript is planned to by published in a Themed Issue in Honour of Professor Vladimir Uversky on the Occasion of His 60th Birthday

Author Response

It would be useful to give a short description of the AlphaFold and TAPASS programs.

Response: We added short descriptions in the Introduction (lines 85-90, 92-95).

It would be good to describe the Figures 2, 3, 4, and 6 in more detail.

Response: More details were added in legends for Figures 2, 3, 4 and 6.

The text contains only one reference to the work by Prof. Uversky, while the manuscript is planned to by published in a Themed Issue in Honour of Professor Vladimir Uversky on the Occasion of His 60th Birthday

Response: Indeed, in several places of the manuscript we describe phenomena, conclusions and findings, which are associated with the studies of Professor Uversky.  Three more references to the work by Prof. Uversky [21, 42, 45] were added.

Reviewer 2 Report

This paper applies bioinformatic methods to analyze differences between canonical and non-canonical isoforms in several large datasets. Using sequence-based predictors from TAPASS they show differences in membrane and extracellular localization and tandem repeat length; and no differences in unstructured regions or aggregation propensity.

They also apply AlphaFold models to look at the details of how isoforms alter protein structure. The impact of splicing on structure is one of the most important questions regarding isoforms. This analysis would have been impossible or extremely expensive to address experimentally, and it is exciting to see AF2 being applied in this way. Furthermore, the "mosaic" nature of isoform variation should be well suited to AlphaFold predictions (given the sensible quality checks the authors used).

Major Critiques

1. Most of the comparative analysis relies critically on the annotation of the canonical isoform in UniProt. The introduction lists a few criteria that depositors consider when deciding which isoform is canonical, but notes that "some of them are still poorly annotated." Are there any studies that quantify the quality of the canonical choice, particularly since you are using UniProt rather than SwissProt?

  I think reliance on the canonical annotation is a limitation of this study and should be clearly addressed. Several of the results investigated are explicitly used in determining the canonical protein (total exon length and expression levels, for instance), making the conclusions depend circularly on the assumptions. One could potentially use metrics that treat all isoforms symmetrically (e.g absolute values of differences), but this would require re-analysis and different statistical models. At a minimum the issue should be brought up in the text.

2. The AlphaFold analysis is very interesting but is mostly qualitative. Would it be possible to provide statistics on what fraction of the structural differences fall into each of the categories? Although the dataset is fairly small it might still provide some quantitative patterns.

3. Additional supplemental data should be provided to allow reproduction of the analysis.
   - List of all Uniprot IDs for the main dataset
   - TAPASS results for each protein
   - Analysis scripts

4. S2 and S3 contain data from ISOexpresso, but more information about how the dataset was processed should be included. Only 82 canonical proteins are included, not 94 as stated in the text. The sets are slightly different as well (Q8NI32-1 in cancer vs Q14667-1 in normal). Finally, how were missing values interpolated?

5. Were the expression levels in cancer or normal patients used for the EAR analysis? Cancer can dramatically change the expression level, but this does not seem to be included in the analysis in section 3.4. It would also be interesting to look at differential splicing in cancer and normal tissues and see if this correlates with any of your predictors.

6. (Line 208) The relationship between expression level and aggregation potential is unclear. Is "aggregation potential" the indicator variable for one of the methods predicting an EAR? I would also rephrase the "Pearson test" sentence, as it convolves the Pearson correlation coefficient (correlation between continuous variables) with the Pearson chi-squared test (which is only valid for categorical outcomes; a t-test should be used here since expression is continuous). A figure should be added showing the distribution of expression levels.

7. Section 3.5 seems contradictory. Fig 6 shows that canonical proteins have more degron motifs (is this the main dataset or the cancer dataset?), but line 303 states the opposite. The link to aggregation potential is also not clear to me.

8. The text should address the possibility that some AF2 models might be incorrect. This is particularly relevant for destabilizing deletions (section 3.7.4), since a low pLDDT can either indicate an unstructured region or simply a region where AlphaFold failed to predict the structure.

Minor Critiques

9. (Section 3.3) Nuclear localization signals are mentioned but no results are shown.

10. (Line 271) Citation needed for "canonical proteins have higher expression levels"

11. Some figures lack statistical significance marks (Fig 5, 6)

12. Section 3.7.1 discusses "non-globular" proteins with tandem repeats, but the example given look globular to me (meaning they form compact structures with hydrophobic cores). Perhaps it would be better to just discuss splice variants in proteins with and without TRs, rather than include globularity.

13. The introduction would benefit from more discussion of the strengths and limitations of AlphaFold.

Author Response

Comments and Suggestions for Authors

This paper applies bioinformatic methods to analyze differences between canonical and non-canonical isoforms in several large datasets. Using sequence-based predictors from TAPASS they show differences in membrane and extracellular localization and tandem repeat length; and no differences in unstructured regions or aggregation propensity.

They also apply AlphaFold models to look at the details of how isoforms alter protein structure. The impact of splicing on structure is one of the most important questions regarding isoforms. This analysis would have been impossible or extremely expensive to address experimentally, and it is exciting to see AF2 being applied in this way. Furthermore, the "mosaic" nature of isoform variation should be well suited to AlphaFold predictions (given the sensible quality checks the authors used).

Major Critiques

  1. Most of the comparative analysis relies critically on the annotation of the canonical isoform in UniProt. The introduction lists a few criteria that depositors consider when deciding which isoform is canonical, but notes that "some of them are still poorly annotated." Are there any studies that quantify the quality of the canonical choice, particularly since you are using UniProt rather than SwissProt?

    I think reliance on the canonical annotation is a limitation of this study and should be clearly addressed. Several of the results investigated are explicitly used in determining the canonical protein (total exon length and expression levels, for instance), making the conclusions depend circularly on the assumptions. One could potentially use metrics that treat all isoforms symmetrically (e.g absolute values of differences), but this would require re-analysis and different statistical models. At a minimum the issue should be brought up in the text.

Response: We agree with this comment. Indeed, construction of properly divided large datasets of canonical proteins and their isoforms represents a challenge because some proteins are still poorly annotated. As far as we know, today, there are no a clear metrics that allows to quantify the quality of the canonical choice. We added in the text (lines 106-107, 112-116) a phrase about limitations of the determination of canonical proteins and explained the Uniprot criteria in more details.

  1. The AlphaFold analysis is very interesting but is mostly qualitative. Would it be possible to provide statistics on what fraction of the structural differences fall into each of the categories? Although the dataset is fairly small it might still provide some quantitative patterns.

Response: We added information about what fraction of the structural differences fall into each of the categories in the text. We also added details in the description of the categories (lines 423, 431, 447-448, 456, 483, 492).

  1. Additional supplemental data should be provided to allow reproduction of the analysis.
    - List of all Uniprot IDs for the main dataset
    - TAPASS results for each protein
       - Analysis scripts

List of UniProt IDs for the canonical and isoform main datasets, analysis scripts and TAPASS results for each protein by prediction tool were included in Supplementary material S2, S3, S4.

4. S2 and S3 contain data from ISOexpresso, but more information about how the dataset was processed should be included. Only 82 canonical proteins are included, not 94 as stated in the text. The sets are slightly different as well (Q8NI32-1 in cancer vs Q14667-1 in normal). Finally, how were missing values interpolated?

Response: We thank the reviewer for pointing on these errors in the description of the datasets. Indeed, we have 82 not 94 canonical proteins in our Supplementary material S5, S6. In the set of isoforms we corrected data related to the isoforms Q8NI32-1 in cancer vs Q14667-1 in normal and removed one of them. As a result, the number of isoform in the dataset became 166 not 167. We corrected these numbers in the main text (lines 133, 317). About the missing values of expression. We kept for the comparison only pairs of canonical proteins with the known expression level in both normal and cancer cells ignoring the missing values of expression.   More information about how the dataset was constructed added to Supplementary Data S5 and S6.

  1. Were the expression levels in cancer or normal patients used for the EAR analysis? Cancer can dramatically change the expression level, but this does not seem to be included in the analysis in section 3.4. It would also be interesting to look at differential splicing in cancer and normal tissues and see if this correlates with any of your predictors.

Response: We agree that the use of expression levels of cancer and normal cells for the EAR analysis was not clearly shown. In the revised version, we substituted Fig 5 with the plots having more detailed information about this relationship and added text (lines 321-342).

  1. (Line 208) The relationship between expression level and aggregation potential is unclear. Is "aggregation potential" the indicator variable for one of the methods predicting an EAR? I would also rephrase the "Pearson test" sentence, as it convolves the Pearson correlation coefficient (correlation between continuous variables) with the Pearson chi-squared test (which is only valid for categorical outcomes; a t-test should be used here since expression is continuous). A figure should be added showing the distribution of expression levels.

Response: The aggregation potential is used for all three methods predicting EARs. To clarify this notion we added text in (lines 309-311). We agree with reviewer about the Pearson test comment.  We performed T-test for the analysis shown in Figure 5).

  1. Section 3.5 seems contradictory. Fig 6 shows that canonical proteins have more degron motifs (is this the main dataset or the cancer dataset?), but line 303 states the opposite. The link to aggregation potential is also not clear to me.

Response: Figure 6 illustrates the results of the main dataset. We explained it now in the legend.

We also corrected the phrase on (lines 361-367) where we inversed “canonical” and “isoforms”. This was the reason of the contradictory statement. We also modified the phrase about the link to aggregation potential.

  1. The text should address the possibility that some AF2 models might be incorrect. This is particularly relevant for destabilizing deletions (section 3.7.4), since a low pLDDT can either indicate an unstructured region or simply a region where AlphaFold failed to predict the structure.

Response:  We added a section 3.7.5 “Limitations of AlphaFold in the interpretation of the conformational changes.”

Minor Critiques

  1. (Section 3.3) Nuclear localization signals are mentioned but no results are shown.

Response:  The result of analysis of Nuclear localization signals was included now in section 3.3. (lines 266-268) and Figure 3C. It was found that the isoforms have remarkably high proportion of the nuclear localization signals in comparison with canonical proteins.

  1. (Line 271) Citation needed for "canonical proteins have higher expression levels"

Response: The references [ref 7, 47] were provided for the indicated expression level.

11. Some figures lack statistical significance marks (Fig 5, 6)

Response: The lacking statistical significance marks were added in Figures 5 , 6.

12. Section 3.7.1 discusses "non-globular" proteins with tandem repeats, but the example given look globular to me (meaning they form compact structures with hydrophobic cores). Perhaps it would be better to just discuss splice variants in proteins with and without TRs, rather than include globularity.

Response: We removed “non-globular” from the Section 3.7.1 title. 

13. The introduction would benefit from more discussion of the strengths and limitations of AlphaFold.

Response:  We added details about the strengths of AlphaFold in the introduction and a special section 3.7.5 about limitations of AlphaFold in the interpretation of the conformational changes (see also comment/response 8).

Round 2

Reviewer 2 Report

All of my concerns have been addressed. Thank you for the additional supplemental scripts and data. I have a few very minor suggestions regarding the corrections:

1. Double check that S5 and S6 are correctly labelled. The files are swapped compared to the original submission and the 'normal' file appears to contain cancer columns like ACC.

2. (Line 211) Check formatting of p-value thresholds